# Features of the Italian Large Dams and their upstream catchments

Giulia Evangelista[1], Paola Mazzoglio[1], Daniele Ganora[1], Francesca Pianigiani[2], Pierluigi Claps[1]

[1] Department of Environment, Land and Infrastructure Engineering, Politecnico di Torino, Torino, 10129, Italy
[2] General Department of Dams and Hydro-Electrical Infrastructures, Roma, 00161, Italy

*Correspondence to*: Giulia Evangelista (giulia.evangelista@polito.it)

**Abstract**

In Italy, a complete and updated database including the most relevant structural information regarding reservoirs and the characteristics of their upstream watersheds is currently missing. This paper tackles this gap by presenting the first comprehensive dataset of 528 Large Dams in Italy. Alongside structural details of the dams, such as coordinates, reservoir

surface area and volume, the dataset also encompasses a range of geomorphological, climatological, extreme rainfall, land cover and soil-related attributes of their upstream catchments. The data used to create this dataset are partially sourced from the General Department of Dams and Hydro-Electrical Infrastructures, as well as from the processing of updated and standardized grid data. These include a digital elevation model from the Shuttle Radar Topography Mission, national-scale monthly maps of hydrologic budget components, land cover and vegetation indices data from the Copernicus Land

Monitoring Service, and high-resolution maps of soil particle size fractions. This dataset (Evangelista et al., 2024a. Available at: https://zenodo.org/doi/10.5281/zenodo.12818297), which contains information not easily available in other similar global or national data collections, is expected to be of great help for a broad spectrum of hydrological applications, particularly those related to floods.

## 1 Introduction

Dams play an important role in managing water resources (Ehsani et al., 2017). The water storage through reservoirs becomes strategic to secure water resources for cities, agriculture and industrial activities during the most critical periods, when natural conditions are inadequate to meet human water needs (Liu et al., 2018). Reservoirs' capacity to retain water volumes can also aid in mitigating downstream flood effects during severe meteorological events (Zhao et al., 2020;

Boulange et al., 2021).

On the other hand, dams can significantly impact the natural flow regime of rivers (e.g. Tundisi, 2018, Barbarossa et al., 2020, Parasiewicz et al., 2023), and these effects may vary depending on the operational procedures employed (e.g. Annys et al., 2020) and the management practices at the catchment scale (see e.g. Kondolf et al., 2018). The way dams are managed can lead to rapid and un-natural flow fluctuations, with consequences on downstream ecosystems (Greimel et

al., 2018). Additionally, an improper management of spillways can result in increased peak discharges in the downstream areas during flood events (Liu et al., 2017).

The proper management and maintenance of dams become crucial to ensure their safe and efficient operation while minimizing any potential negative impacts, particularly as many of these structures, built in the first half of the twentieth century, are now aging. This involves addressing the complex inter-relationships between dams and their host

environment. To this aim, knowing the exact position of reservoirs, as well as their age, primary use, structural features, and the main characteristics of their upstream basins can help researchers and decision-makers better understand the potential risks and benefits associated with their operation on whatever large basin scale (Speckhann et al., 2021).

In recent years, some research works have been conducted to obtain global-scale collections of data on these infrastructures; such initiatives aim to catalog large dams worldwide, facilitating cross-country comparisons.

The first global-scale dataset was the Global Reservoir and Dam database (GRanD) (Lehner et al., 2011), developed within the Global Water System Project. GRanD contains information regarding more than 6800 dams and their associated reservoirs.

Later on, the GlObal geOreferenced Database of Dams (GOODD) (Mulligan et al., 2020) was released. GOODD is a global dataset of more than 38,000 georeferenced dams containing both their geographic coordinates and information on the associated catchment areas. Dams were digitized by scanning tiles on the Google Earth geobrowser (https://earth.google.com/web/), using a GeoWiki coded in KML (Keyhole Markup Language). Catchment boundaries were derived from the HydroSHEDS SRTM-based Digital Elevation Model (Lehner et al., 2008) at 30-arc second resolution for the latitude from 60° North to 60° South, complemented with the Hydro1K DEM at 30-arc second resolution for the remaining 30° amplitude bands at the poles. The GOODD and GRanD datasets are among those that constitute the recent river barrier and reservoir database developed under the Global Dam Watch (GDW) initiative (Lehner et al., 2024), which collects data on over 40,000 dam points worldwide.

Recently, Zhang and Gu (2023) released the GDAT (Global Dam Tracker) dataset, containing more than 35.000 reservoirs all over the world, where dam coordinates are verified using geospatial software and catchment areas are derived from satellite data products. At the continental level, a geo-referenced collection of artificial instream barriers in 36 European countries was compiled as part of the AMBER (Adaptive Management of Barriers in European Rivers) Project (Belletti et al., 2020). Additionally, the Dataset of georeferenced Dams in South America (DDSA) was published in 2021 (Paredes-Beltran et al., 2021).

Unfortunately, the spatial coverage of these global/continental datasets on a national scale, for example in Italy, is rather low. When considering Italy, only 144 and 87 dams are collected in the Global Dam Tracker and the Global Reservoir and Dam database, respectively. The GlObal geOreferenced Database of Dams includes a total of 245 large dams in Italy, defined as those higher than 15 meters or with a storage volume exceeding 1,000,000 cubic meters. However, it is worth noting that these 245 dams represent only about half of all large dams in Italy. The recently released GDW dataset comprises 313 points in Italy. However, for many of them, several variables that should be part of the dataset are not available.

These global-scale collections might not always be suitable for national- and regional-scale investigations, mainly due to data resolution or different definitions and criteria used by different countries to categorize dams, which can lead to inconsistencies when integrating data from global sources into local investigations. Furthermore, national and regional investigations often benefit from the input of local experts and authorities who possess in-depth knowledge of specific dams and reservoirs. Global datasets may not always have access to this localized information. National-scale datasets, like the one presented in this work, would be of great benefit for complementing or providing updates to existing continental or global datasets. National registers and inventories are available in countries like India (National Register of Large Dams, available at https://cwc.gov.in/national-register-large-dams), the UK (available at https://www.data.gov.uk/dataset/aa1e16e8-eded-4a60-8d1d-0df920c319b6/inventory-of-reservoirs-amounting-to-90-of-total-uk-storage) and the US (National Inventory of Dams, NID, available at https://nid.usace.army.mil/#/), among others. Shen et al. (2023) have recently released a dataset of 3254 Chinese reservoirs that also contains landscape attributes of the upstream watersheds and related hydrometeorological time series, while Speckhann et al. (2021) produced an inventory of 530 dams in Germany with information on name, location, river, start year of construction and operation, crest length, dam height, lake area, lake volume, purpose, dam structure, and building characteristics.

In Italy, as of today, a complete national-scale open access inventory of large dams that also includes key characteristics of upstream catchments, needed for hydrological studies, is not available. The General Department of Dams and Hydro-

Electrical Infrastructures (referred to GDD hereinafter) recently published a digital cartography of Italian Large Dams (available at https://dgdighe.mit.gov.it/categoria/articolo/_cartografie_e_dati/_cartografie/cartografia_dighe, in Italian), which provides some general information on the use and the structure of the dams, together with some geometrical features. In particular, the dam's height, the storage volume and the elevation of the spillway crest are given. However, information on the upstream basins is missing.

Similarly, the Istituto Superiore per la Ricerca e la Protezione Ambientale (ISPRA), in its report on water resources in Italy (Policicchio, 2020), provides general information and some structural characteristics of the Italian dams, but it still lacks a characterization of the upstream basins, as the only information given is their drainage area, with no coverage of the watershed boundaries.

In this work we present a comprehensive collection of the features of 528 Large Italian Dams and related catchments. In section 2, we provide information on dams' classification, type and purpose, as well as the years of start and end of construction works, while in section 3 their structural characteristics are reported. In section 4 geomorphological, climatic and soil-related attributes of the upstream catchments are described. In section 5 some useful elements are suggested for an expeditious assessment of the interaction of dams and their host environment, and some conclusions are finally drawn in section 6.

## 2 Classification and role of reservoirs over the Italian territory

In Italy, the General Department of Dams and Hydro-Electrical Infrastructures (GDD) currently oversees 528 Large Dams, which can be classified according to their function, i.e. the specific purposes they serve, and their construction typology, which pertains to structural aspects. An overall picture of the Italian reservoir system is given in Figure 1, where both their geographical distribution throughout the country (Figure 1a) and their grouping by classes (Figures 1b and 1c) are shown. The primary function of more than half of the Italian reservoirs, mainly concentrated in the Alpine region, is hydropower generation. Additionally, around 10% of the dams serve irrigation purposes, particularly in the central and southern areas, while a smaller percentage was designed for flood control, industrial use or drinking water supply. A consistent portion of dams in Italy serve multiple functions, combining two or more of the purposes mentioned above.

Currently, about 20% of the reservoirs are temporarily out of service (black color in Figure 1a), with some no longer storing water, undergoing functional and technical tests (gray color in Figure 1a) or under construction (white color in Figure 1a). Southern Italy hosts the largest concentration of dams undergoing testing.

Approximately 8% of the 528 reservoirs supervised by the GDD are classified as river barrages, marked with triangles in Figure 1a. According to the technical literature, here the term "river barrage" refers to a structure designed to create a contained backflow within the riverbed. Its primary purpose is to raise the upstream water level to enable water diversion and, more broadly, to regulate water levels. Consequently, a river barrage is not typically intended for water storage. Some of these structures are located at large natural lakes, regulating water outflows, ensuring the stability of water levels, and supporting specific water management objectives. Some examples include the Miorina barrage (which controls the water outflows from the Maggiore Lake); Olginate (which regulates the Adda river and the water level of the Como Lake); Salionze (which manages the water level of the Garda Lake); and Sarnico (that controls the water release from the Iseo Lake).

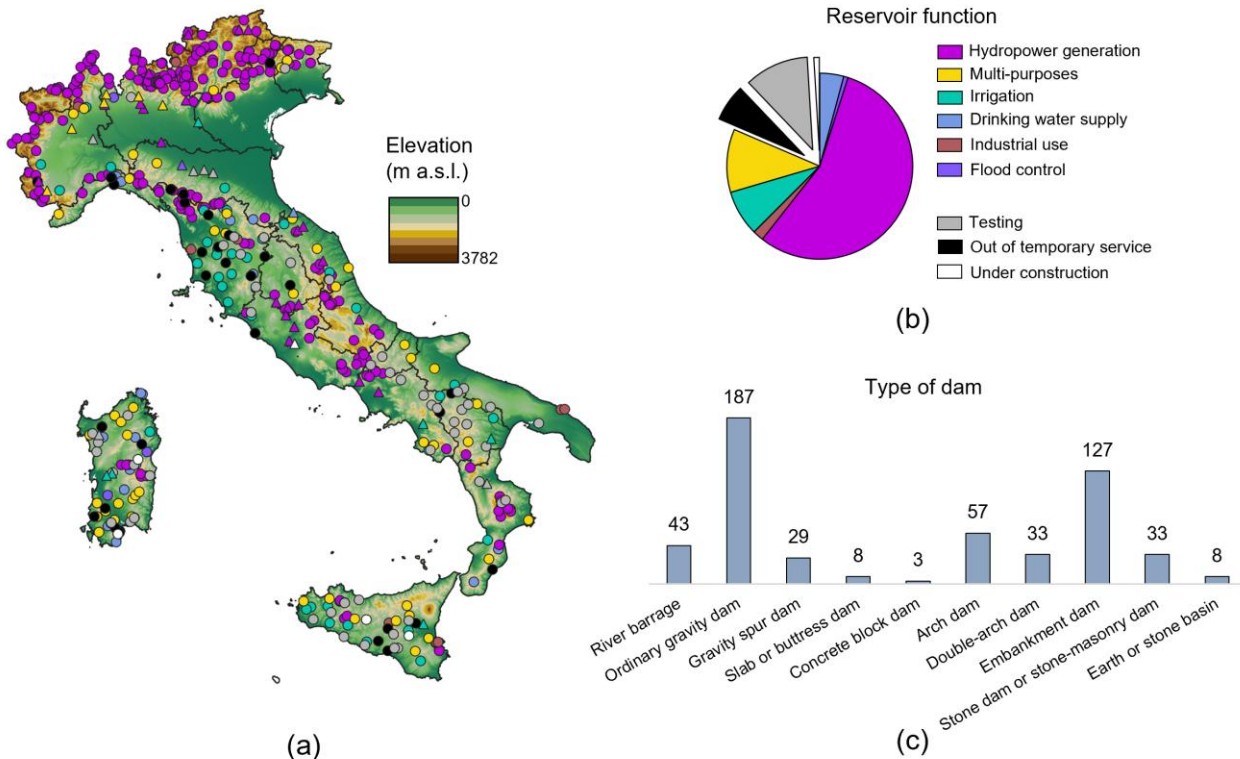

**Figure 1. Spatial distribution of the 528 Italian Large Dams (a), with an overview of their functions (b) and construction types (c). Triangles in panel (a) correspond to river barrages, while all the other dams are identified by circles.**

In this paper, all the characteristics discussed above, along with additional information, such as the coordinates of the dam and the years of the start and end of construction, are listed in Table 1 and are available for each dam. The geographical coordinates, sourced from the GDD, have been carefully checked manually using GIS software, in order to verify that they actually matched the dam wall at the "meter" accuracy.

**Table 1. Descriptive features of the dams.**

| Parameter | Notation | Description |
|---|---|---|
| Name | dam | - |
| x coordinate | dam_xcoord | x coordinate of the dam expressed in the reference system WGS84/UTM 32N (EPSG 32632) |
| y coordinate | dam_ycoord | y coordinate of the dam expressed in the reference system WGS84/UTM 32N (EPSG 32632) |
| Technical Department | technical_dep | Administrative and operational sub-structure responsible for managing the dam |
| Region | region | - |
| Province | province | - |
| Starting year of construction | y_start | - |
| Ending year of construction | y_end | - |
| River | river | Dammed watercourse |
| Affected river | river_release | Rivers affected by water releases from the dam |
| Purpose | purpose | Drinking water supply, Flood control, Hydropower generation, Industrial use, Irrigation, Multi-purposes |
| Status | status | Limited reservoir, Normal service, Out of temporary service, Testing, Under construction |
| Construction type | building | Arch dam, Concrete block dam, Double-arch dam, Earth or stone basin, Embankment dam, Gravity spur dam (either full or with |

internal compartments), Ordinary gravity dam (either concrete or masonry), River barrage (either concrete or masonry), Slab or buttress dam, Stone or stone-masonry dam

A significant portion of the large dams in Italy dates back to the central part of the 20th century, as shown in Figure 2, which illustrates the total number of dams built in each year from 1870 to 2010 (represented by points) and the year each dam was completed (represented by lines). The influence of the World War II is clearly evident in a breakdown in dam construction (red arrow in Figure 2), while the sharpest growth rate occurred between 1950 and 1970. Knowing the temporal distribution of dams and, particularly, the end year of construction, is crucial for conducting accurate flood frequency analyses (e.g. Villarini et al., 2011; López and Francés, 2013), as well as for calibrating and updating hydrological models (e.g. Chaudhari and Pokhrel, 2022).

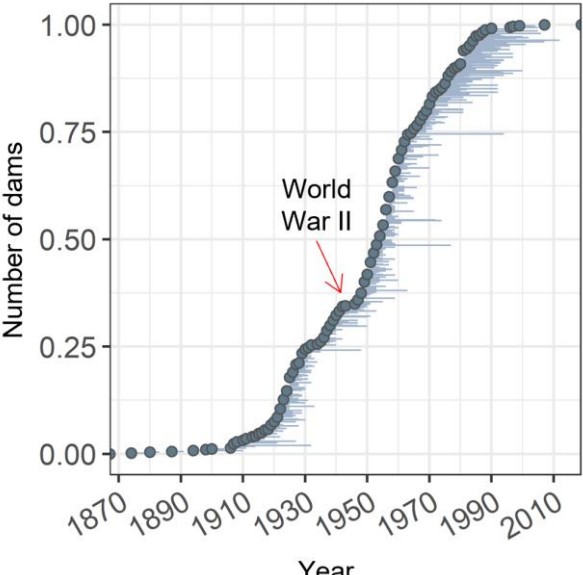

**Figure 2. Total number of dams, expressed as a percentage, whose construction started in each year between 1870 and 2010 (points), year of completion of each dam's construction (lines). The red arrow highlights a disruption in dam construction during the World War II. Dams currently under construction or lacking available start and end construction dates have not been included.**

Some of the 528 structures serve as secondary dams, i.e. additional dam structures built within a single lake system. These are shown in Table A.1 of Appendix A. In this paper, the term "secondary" refers to a dam that is smaller in size than the main one.

## 3 Structural features of the dams

Information about the main structural features of the Italian dams has been obtained from the GDD. An overview of these characteristics, together with a summary description, is available in Table 2, while Figure 3 shows the spatial distribution and variability of storage volumes and lake areas all over Italy. While dam height and storage volume are easily accessible data, what distinguishes this dataset from similar ones is first the inclusion of the lake area. Although this information can be found in the GDW dataset (Lehner et al., 2024), it should be recalled that there are only about 60 % of Italy's large dams represented in that collection. Furthermore, the elevation above sea level corresponding to the reported lake area is not clearly specified. Information on lake area is crucial when conducting assessments of the dam's ability to mitigate flood peaks, as the reservoir surface area exerts a direct influence on the dam's capacity to manage excess water during

periods of high flow. This parameter is directly involved, for instance, in the computation of the Synthetic Flood Attenuation Index (SFA), developed by Miotto et al. (2007). In a more recent study, Cipollini et al. (2022) introduced an index that quantifies the impact of a reservoir in mitigating flood peaks and that relies on the following parameters: i) the area of the upstream catchment; ii) the lake area; iii) the spillway length; iv) the slope $n$ of the Intensity-Duration-Frequency rainfall curve (see Eq. (2) in section 4.2.4).

The elevation corresponding to the maximum allowed water level is another information one should not overlook. By subtracting this elevation by the elevation of the spillway crest, one obtain a metric that, when multiplied by the lake area, directly provides an estimate of the volume available for flood mitigation (Eq. 1 in Table 2). Furthermore, the wider the gap between these two elevations, the greater the potential for using the reservoir for flood mitigation while retaining the whole volume for hydropower generation or storage of irrigation supplies. Data related to the elevation of the maximum allowed water level is rarely made available in the already released databases.

In order to ensure the accuracy and reliability of the lake area measures, a careful validation has been undertaken. This has involved a systematic comparison of the values retrieved from the GDD with those acquired from a high-resolution Digital Elevation Model, namely the TINITALY/01 DEM at 10 m resolution (Tarquini et al., 2007), at the elevation corresponding to the spillway crests. In case of major differences, the lake area values have been corrected, also by checking the dam design reports. It must be specified that in the case of river barrages the lake area is not provided, as it cannot be unequivocally identified. For dams currently out of temporary service (indicated by black dots or triangles in Figure 1a) a null lake area has been assigned.

It must be specified that the structural characteristics of the dams are presented without uncertainty information, as the data was sourced from official, controlled sources, which minimizes the need for additional uncertainty assessments.

**Table 2. Structural features of dams.**

| Parameter | Notation | Units | Description |
|---|---|---|---|
| Height | H | m a.s.l. | Height of the dam wall. |
| Elevation of the spillway crest | H_s | m a.s.l. | If multiple spillways are located at different elevations, the highest one has been considered. |
| Elevation of the maximum allowed water level | H_all | m a.s.l. | Highest elevation at which water can be stored in a reservoir without overtopping the dam. |
| Elevation of the dam crest | H_c | m a.s.l. | Elevation of the top of the dam. |
| Reservoir storage volume | V_s | Mm$^3$ | Volume measured at the elevation H_s, according to Law No. 584 of 21$^{st}$ October 1994. It is defined as the capacity enclosed between the highest elevation of the spillways, or the top of any gates, and the elevation of the lowest point on the upstream face of the dam wall. |
| Reservoir volume available for flood attenuation | V_f | Mm$^3$ | Volume computed as $(H\_all - H\_s) \cdot A\_l$ (1) |
| Lake area | A_l | km$^2$ | Lake area measured at the elevation H_s. |

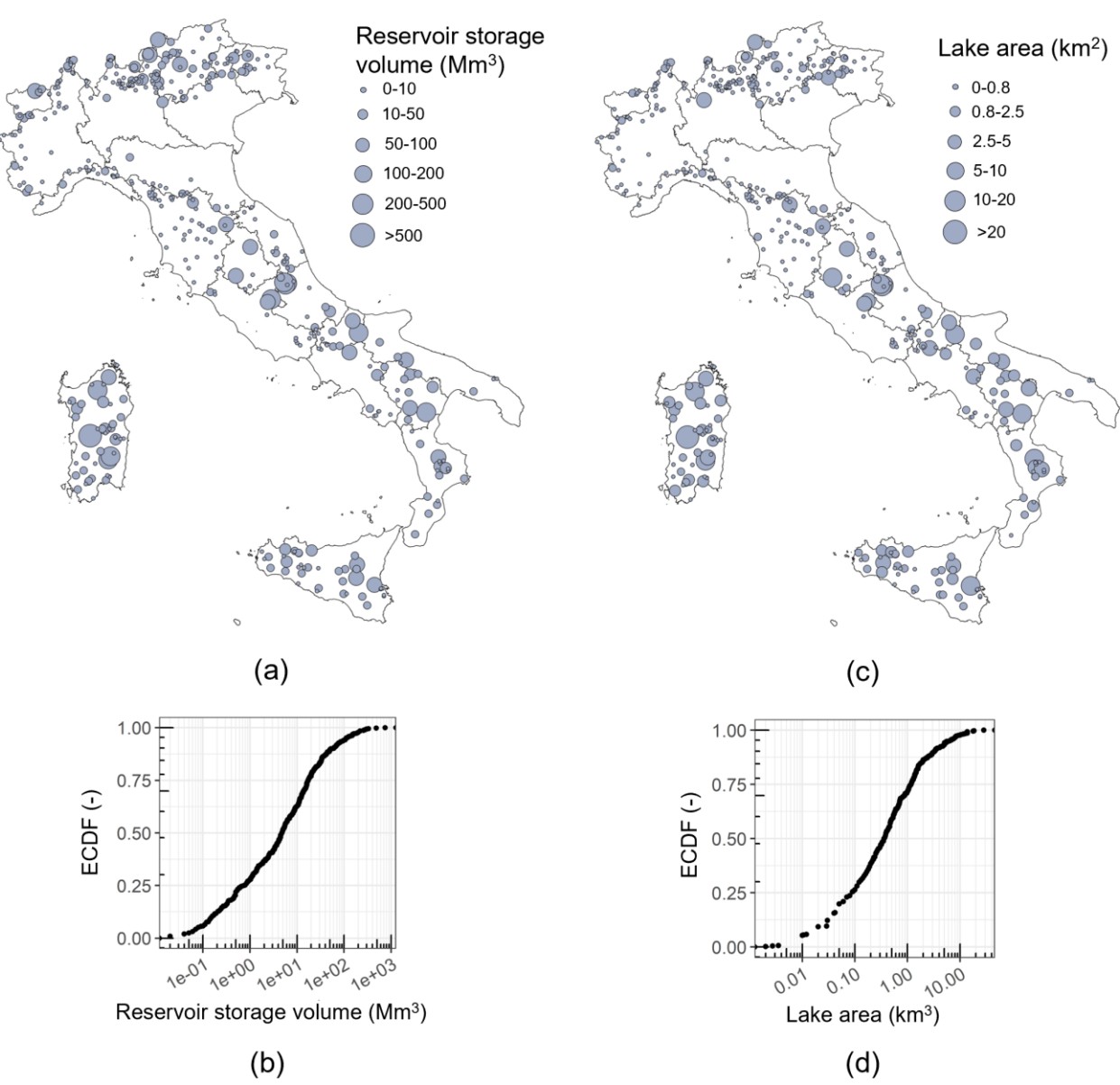

**Figure 3. Variability, all over Italy, of storage volumes (V_s) and lake areas (A_l): spatial distributions (a, c), empirical cumulative distribution functions (ECDFs) (b, d). The ECDF is defined as $\frac{i}{N}$ for *i*=1,..., *N* where *i* is the ordered variable for each reservoir. River barrages and temporarily out-of-service dams have not been included.**

Figure 4 illustrates the relationship between reservoir storage volumes and their surface areas. A linear relationship in a log-log graph is found, and the corresponding power-law fitting equation is provided. The relationship depicted in Figure 4 holds a practical value as it shows that, in cases where the lake area is not known, one can still approximate it using the reservoir storage volume as a good proxy.

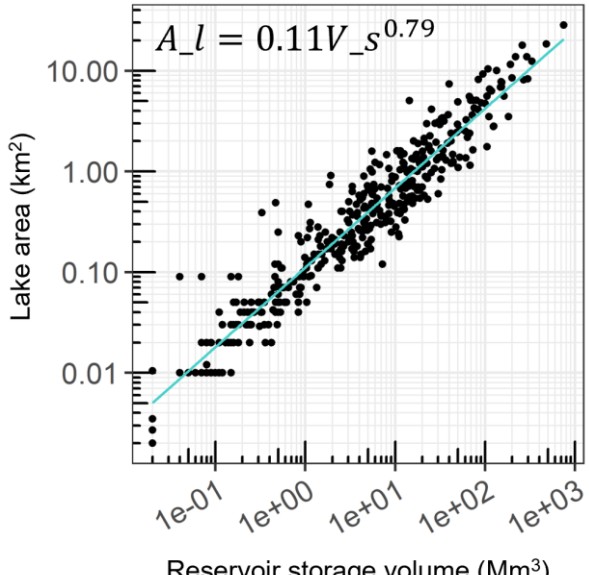

$$A\_l = 0.11 V\_s^{0.79}$$

**Figure 4. Relationship between reservoir volumes (V_s) and their surface areas (A_l). River barrages, secondary dams and temporarily out-of-service dams have not been included to fit the equation shown in the Figure.**

**4 Morphological and climatic characterization of the upstream catchments**

Italy, like many other regions globally, is experiencing shifts in climatic patterns due to climate change (van Vliet et al., 2015; Bombelli et al., 2019). The incumbent increasing frequency and intensity of extreme events (Libertino et al., 2019; Mazzoglio et al., 2022) poses new challenges for dam safety. Historical flood data may no longer represent the full range of potential flood events, and this would entail a re-assessment of spillway design flood to accommodate changing hydrological conditions, as already stated ten years ago by Bocchiola and Rosso (2014). The insufficient knowledge of

basic hydrological information in areas where dams are located, in view of the need for flood attenuation plans and re-assessment of dam's hydraulic safety, has been also stressed by the Italian Committee on Large Dams (ITCOLD, 2023). Accurate and up-to-date information about the catchment's hydrological response is then crucial for reconsidering the potential flood scenarios that the dam may face. Furthermore, in 2018 the GDD proceeded to update the directives established in 2000 (Directives number SDI/7128 and SDI/8111, 2000), specifically focusing on the reconstruction of

incoming         hydrographs         (Directive         number         3356,         2018.         Retrieved         from: https://www.dighe.eu/normativa/allegati/2018_Circ_DGDighe_13-02_n_3356.pdf). These updated regulations now mandate dam managers and concessionaires to reconstruct the most severe hydrological event of the year, in addition to one or more significant events from the previous five years (Santoro et al., 2023). Within this framework, factors such as catchment size, shape, slope, and land cover play a significant role in allowing to calibrate models for the estimation of

the flood hydrograph and its peak flow.
       The following sections provide a description of the process used to delineate the basin boundaries, along with an overview of the computed attributes for each catchment. The list of attributes, as well as the rationale and the methodologies adopted, are the same as that presented in Claps et al. (2024), where 631 gauged watersheds have been characterized from a climatic, soil, and geomorphological point of view. The choice to provide the same basin attributes moves from the

possibility of increasing the hydrological knowledge throughout Italy, almost doubling the number of basins where the same level of information is available.

**4.1 Catchments boundaries**

The boundaries of the upstream catchments were computed by processing the Shuttle Radar Topography Mission (STRM) Digital Elevation Model at 30 m spatial resolution (Farr et al., 2007). The selection of the SRTM DEM, despite its coarser resolution compared to TINITALY/01 (Tarquini et al., 2007) or TINITALY/1.1. (Tarquini et al., 2023) is based on considerations on how the TINITALY DEMs are compiled. These latter ones are composite DEMs created by merging different regional or local DEMs, which means they do not maintain a consistent level of accuracy nationwide. To address this issue, we opted to use the SRTM DEM.

The processing of the DEM has been carried out using the *r.basin* GRASS GIS add-on (Di Leo and Di Stefano, 2013), following the procedure described in Figure 1 in Claps et al. (2024). The process of establishing basin boundaries includes calculating drainage directions, determining flow accumulation, and finally extracting the stream network with a specified threshold value for channel initiation. Here a minimum area of 0.1 km$^2$ has been adopted to extract basins larger than 1 km$^2$, otherwise a threshold of 0.02 km$^2$ has been used.

As expected, the automatic delineation procedure did not exhibit major problems, since the basins analyzed are typically located in mountainous areas, where elevation differences are quite pronounced. In a couple of cases, where the topography posed difficulties for automated delineation, basin masks were created manually, i.e. forcing the DEM to correct the Total Contributing Area (TCA) map built by the *r.basin* procedure, a technique commonly referred to as *stream-burning* (Lindsay, 2016). When needed, this operation has been conducted by comparing the unconstrained river network generated by *r.basin* with the reference one provided by the Istituto Superiore per la Ricerca e la Protezione Ambientale (ISPRA, available at http://www.sinanet.isprambiente.it/it/sia-ispra/download-mais/reticolo-idrografico/view). Only two watersheds required a *stream-burning*: those upstream of the Panaro dam, in Emilia-Romagna, and the Lago Pusiano dam, in Lombardia. Both of these basins mainly include flat areas, characterized by the presence of several artificial diversions. The Lake Pusiano dam area has also undergone substantial urbanization. On the other hand, five reservoirs, namely the Gerosa dam (Marche), the Presenzano dam (Campania), the Vasca di Edolo (Lombardia), Vasca Ogliastro (Sicilia) and Vasca Sant'Anna (Calabria) dams, do not possess a directly associated catchment area. This is either because they have very little to no upstream contributing area, as in the case of the Gerosa dam, or because they serve as off-stream reservoirs, storing water outside the natural course of a river. Therefore, information regarding the upstream catchments for these dams is unavailable, that reduces the total number of basins collected in this database to 523.

Since the geographical coordinates of the dam do not necessarily coincide with that of the basin outlet, i.e., those operationally used when defining the catchment boundaries, a second pair of coordinates than those listed in Table 1 is provided. They are referred to as "operational coordinates" and named basin_xcoord and basin_ycoord in the dataset.

The reliability of watersheds automatically extracted from a DEM is inherently influenced by several factors. These include the resolution and accuracy of the DEM itself, the precision and robustness of the extraction algorithm, and the handling of specific landscape features such as flat areas and depressions. Errors in the DEM, such as noise or inaccuracies in elevation data, can propagate through the watershed delineation process, leading to potential misrepresentations of watershed boundaries and associated topographic features. Consequently, it is crucial to assess and validate these automatically delineated watersheds against ground truth data or higher-quality references.

As mentioned above, DEM conditioning procedures were only necessary in very few cases in our study. To ensure the accuracy of the basin contours determined using the above-mentioned procedure, some verification steps have been undertaken. In a first instance, Google Earth has been employed. The delineated watershed boundaries have been cross-referenced with the visual representation of the surrounding terrain offered by Google Earth, ensuring that they coincide

with the natural ridges and topographical features of the area. Then the basin area values have been compared with those reported in Policicchio et al. (2020), as will be discussed in section 4.2.1.

## 4.2 Catchment attributes

Derivation of all the catchment boundaries as described in the above section allowed us to calculate several catchment-averaged attributes. The attributes computed are the same as that provided in Claps et al. (2024) and can be grouped in four main categories, as: geomorphological, soil, land cover and NDVI (Normalized Difference Vegetation Index), climatological and attributes related to extreme rainfall. Hence, while not explicitly reported in the main body of the paper, tables listing all the descriptors provided can be found in Appendix B.

It is important to recognize that all the attributes presented in this section are associated with some degree of uncertainty. As mentioned earlier, potential inaccuracies in geomorphological descriptors should be considered in relation to the processing of the digital elevation model. On the other hand, uncertainties in soil, land cover and climatological attributes, which are computed from existing raster datasets, may result from interpolation or spatial resampling procedures. While we will explore the uncertainties related to geomorphological descriptors in more detail below, readers can refer to Claps et al. (2024) for an in-depth discussion on the uncertainties associated with the other descriptor categories.

### 4.2.1 Geomorphological attributes

This category of attributes was computed by using additional algorithms to *r.basin*, i.e. *r.stat*, *r.slope.aspect*, *r.stream.stats* (Jasiewicz, 2021) and *r.accumulate* (Cho, 2020) functions, as described in Claps et al. (2024).

In Figures 5a and 5b, each dot represents a dam, and its color indicates the mean elevation (a) and mean slope (b) of the upstream basin. The point size corresponds to the basin area, normalized with respect to the largest basin within the dataset, i.e., the one upstream of the Isola Serafini river barrage (in between Emilia-Romagna and Lombardia). From the overall picture of the 523 watersheds, one recognizes that the set of basins is composed by mainly small, mountainous basins, as almost 75% of them have a normalized area of less than 0.2, and approximately 50% of them have an average slope of over 20%. Almost 40 dams have an area of less than 1 km$^2$. It must be specified that the analysis focused solely on the watersheds directly connected to the reservoirs, without considering any indirectly connected basins. In other words, areas that are part of the reservoir management system but do not lie within the natural contributing basin upstream of the dam were not considered. In Figure 5c, the variability of empirical hypsometric curves for the 523 basins is depicted, showing rather different morphometric characteristics across the dataset.

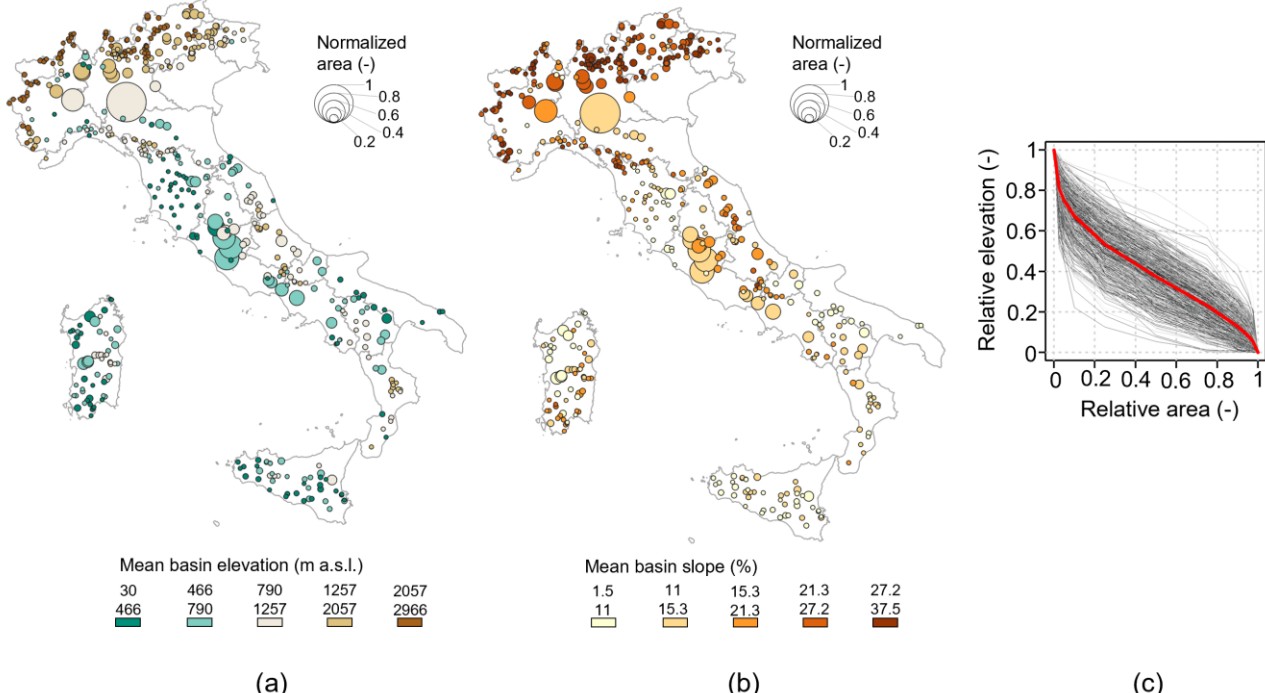

**Figure 5. Mean elevation (a) and mean slope (b) of the 523 basins. The point size represents the basin area normalized to the maximum value within the dataset. Empirical hypsometric curves of each basin (in gray) and average hypsometric curve (in red) (c).**

The only available benchmark for controlling basin areas is a report provided by ISPRA (Policicchio, 2020). Titled "Water Resources in the Geological Context of the Italian Territory: Availability, Large Dams, Geological Risks, Opportunities", this report aims to gather information on water resources, particularly their use in the context of artificial barriers like dams. It also explores the relationship between water resources and various natural hazards, including seismic, tectonic, geomorphological, and hydraulic risks. We have compared the basin area values published in the above report with those determined using the *r.basin* algorithm and the resulting scatter plot is presented in Figure 6a. Notably, the basin areas reported by Policicchio (2020) tend to be higher than those directly computed here, particularly for basins smaller than 100 km$^2$. This discrepancy may be likely attributed to either inaccuracies in outlet coordinate placement or to the inclusion of indirectly connected basins in the computation of the upstream area. It is important to consider that employing a digital elevation model with a coarser resolution (not specified in Policicchio (2020)) may have played a role in the observed discrepancies, exerting a more significant impact on smaller areas. Given that manual adjustments (*stream burning*) were necessary for only two basins out of 523 and considering that the areas were derived directly from DEM processing, it can be concluded that the calculated areas are reasonably accurate. As an additional quality control, the consistency between the length of each basin's main channel and that of its longest drainage path has been assessed. As detailed in Claps et al. (2024), this procedure aims to identify potential weaknesses in the delineation procedure. Ad-hoc checks highlighted occasional issues in the GIS procedure for computing the main channel length and the longest drainage path length. Two problems were observed: (i) main channel shapefiles contained multiple features that required merging, and (ii) multiple longest drainage paths were identified for the same catchment, each differing by no more than 100 meters. In this latter case, one longest drainage path was chosen manually, and other instances were removed. After these checks, a linear relationship between the length of the main channel and the basin area has been found in a log-log graph, as shown in Figure 6b and discussed in Claps et al. (2024).

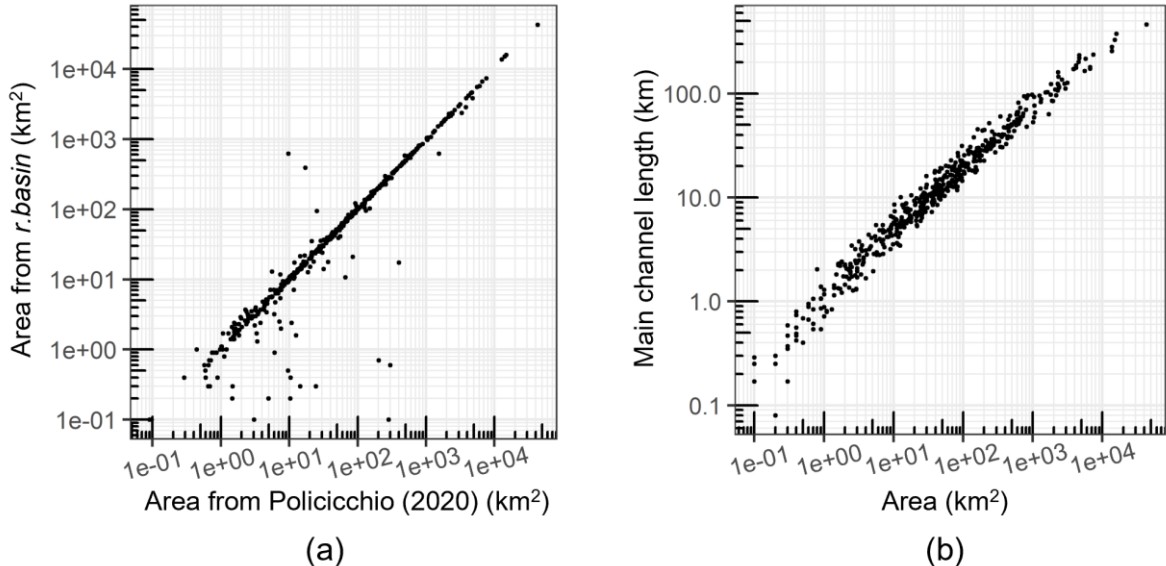

**Figure 6. Quality controls on geomorphological parameters. Comparison of area values obtained from the delimitation procedure and those reported in Policicchio et al. (2020) (a). Relationship between the main channel lengths and the basin areas (b).**

### 4.2.2 Soil, Land cover and NDVI attributes

For each catchment, spatially-averaged information about soil, land cover and NDVI were computed. Similarly to the other categories of attributes, the list of soil, land cover and NDVI attributes provided is analogous to the one reported in Claps et al. (2024) and the reader will find it in Appendix B (Tables B.2 and B.3). The dataset encompasses soil descriptors that offer insights into area-averaged soil permeability conditions, like the Curve Number (Soil Conservation Service, 1972) and the saturated hydraulic conductivity.

The Curve Number is an empirical parameter that assesses the proportion of total rainfall converted into net rainfall during a flood event. Three different values for the Curve Number (CN1, CN2 and CN3) are available, each corresponding to specific antecedent wetness conditions of the soil (i.e., dry, average, or wet condition, respectively). For each of these three Curve Numbers, the spatial mean value and the spatial coefficient of variation were computed. Consistently with Claps et al. (2024), the data source used is the Curve Number national-scale raster maps at 250 m resolution computed by Carriero (2004).

The mean saturated hydraulic conductivity has been computed using the pedo-transfer function proposed by Saxton et al. (1986). For its application, that requires sand and clay content, we employed soil texture fraction values extracted from the SoilGrids maps (Hengl et al., 2017; available at https://soilgrids.org/). These maps delineate soil property parameters at 250 m spatial resolution all over the world across seven standard depths, from 0 cm to 200 cm. These have been computed using over 230,000 soil profile observations from the WoSIS (World Soil Information Service) Database (Batjes et al., 2009). To align the information produced with the hydrological context of this study, we averaged the soil texture information over the initial 30 cm of soil depth, consistently with Claps et al. (2024). This depth range is often regarded as representative of the topsoil layer, essential for supporting vegetation and regulating both water retention and drainage processes.

Land cover characteristics were derived from the 100 m spatial resolution third level of CORINE Land Cover 2018 (available at: https://land.copernicus.eu/). This involved the redistribution of 44 land cover classes into 5 distinct land

indices, i.e.: clc1, associated to urbanized areas; clc2, related to arboreal vegetation; clc3, corresponding to herbaceous vegetation and crops; clc4, associated to non-vegetated and industrial areas; cl5, related to humid areas. For each catchment, we provided the percentage covered by each of these 5 classes.

Data related to the stability of vegetation over time within the basin, in terms of growth, health, and coverage, has been further assessed by means of the NDVI (Normalized Difference Vegetation Index). The NDVI metric provides measures of vegetation density and health, offering information about land-use changes and potential impacts on hydrological processes, specifically on the way water is absorbed and released through plant transpiration. Multi-temporal indicators of the NDVI were computed from the Copernicus Land Monitoring Service (available at https://land.copernicus.eu/) Long Term Statistics (LTS) NDVI V3.0.1, with a spatial resolution of 1 km. This dataset was used to evaluate NDVI mean observations spanning the 1999-2019 period for each of the 36 ten-daily periods, resulting in 36 raster maps. These maps facilitated the computation of: the mean annual NDVI value, the (temporal) coefficient of variation of the NDVI, and the spatio-temporal mean NDVI regime. The latter was synthetically characterized using a Fourier series representation, that allows the description of the shape of the regime with only four parameters. This representation provides a more concise overview than the 36 individual 10-day average values. More details about the computation of the 4 coefficients of the Fourier series are available in Claps et al. (2024).

### 4.2.3. Climatological attributes

National-scale datasets with a resolution of 1 km were employed to assess various climatological attributes that allow to describe the precipitation and temperature climatology of these catchments.

Mean monthly precipitation data were extracted from the BIGBANG 4.0 dataset (Bilancio Idrologico GIS Based a scala Nazionale su Griglia regolare; Braca et al., 2021). Spanning from 1951 to 2019, this dataset applies spatial interpolation at a 1 km resolution to rain gauge measurements. It also incorporates already available spatial interpolations from ArCIS (Archivio Climatologico per l'Italia Centro Settentrionale; Pavan et al., 2019) in limited areas and for specific years. Mean monthly temperature information is similarly derived from this dataset.

Catchment boundaries were used to clip the aforementioned precipitation and temperature maps, enabling the derivation of spatial averages for the 14 climatological attributes.

Similarly to what done for the NDVI averages, both monthly precipitation depths and monthly temperature data are processed to calculate the mean coefficients of the Fourier series, providing an approximation of the precipitation and temperature regimes (four coefficients for rainfall and four for temperature, as described in Claps et al. (2024)). The original monthly raster maps were also used for the evaluation of the temporal coefficient of variation of rainfall regimes and the time lag between maximum and minimum of the mean monthly rainfall.

The same dataset was employed to determine the Mean Annual Precipitation (MAP) and the Mean Annual Temperature (MAT) basin-averaged values, along with their spatial coefficients of variation.

### 4.2.4 Extreme rainfall attributes

According to the index-value approach (Darlymple, 1960), based on the "simple scaling" hypothesis (Burlando et al., 1996), the quantile of the annual maximum rainfall depth for the duration d and the return period T can be expressed by means of the Intensity-Duration-Frequency (IDF) curves, defined as:

$$h(d,T) = K_T \cdot a \cdot d^n \tag{2}$$

where $a$ and $n$ are the scale factor and the scaling exponent, respectively, and $K_T$ is the "non dimensional inverse frequency factor", also called "growth factor". Catchment-averaged values of $a$, $n$, and $K_T$ have been derived on all the considered basins, by interpolating values previously determined at the individual rain gauge level. A complete rain gauge network is available in the Improved Italian – Rainfall Extreme Dataset (I[2]-RED), a collection of short-duration (1, 3, 6, 12 and 24 hours) annual maximum rainfall depths measured by more than 5000 rain gauges from 1916 up to the present (Mazzoglio et al., 2020). The at-site $a$ and $n$ rainfall parameters are obtained by means of linear regression of the logarithm of the average rainfall maxima $h(d)$, computed from at-site measurements over the 1- to 24-hour durations, with the logarithm of the duration $d$. The growth factor $K_T$ can be estimated using the sample L-moments of the time series, after having defined a specific probability distribution. Time series of at least 10 years of data have been used to estimate $a$ and $n$, while 20 and 30 years have been set as minimum lengths for the computation of the coefficients of L-variation (*L-CV*) and L-skewness (*L-CA*), which have been computed using Eqs. (6) and (7) of Laio et al. (2011), respectively.

At-site rainfall statistics have been then spatial interpolated at 250 m resolution with the autokrige R function (Hiemstra and Skoien, 2023). We calculated catchment-averaged values for $a$ and $n$, and the basin mean *L-CV* and *L-CA* coefficients for the durations of 1, 3, 6, 12 and 24 h to compute a catchment-averaged $K_T$.

Again, the list of rainfall attributes provided is analogous to that reported in Claps et al. (2024) and the reader will find it in Table B.5 in Appendix B. Figures 7a and 7b show the outcome of the spatial averaging process applied to parameters a and n, respectively, at the catchment level. It is interesting to note an inverse dependence between $a$ and $n$, i.e. where the $n$ parameter shows high values, indicating a slow decrease in rainfall intensity with duration, the $a$ value, which represents the average maximum rainfall over a one-hour duration, tends to be low. This pattern is particularly evident in Alpine regions and, as shown by Evangelista et al. (2023), contributes in part to justify the significant flood attenuation capacity of Alpine dams.

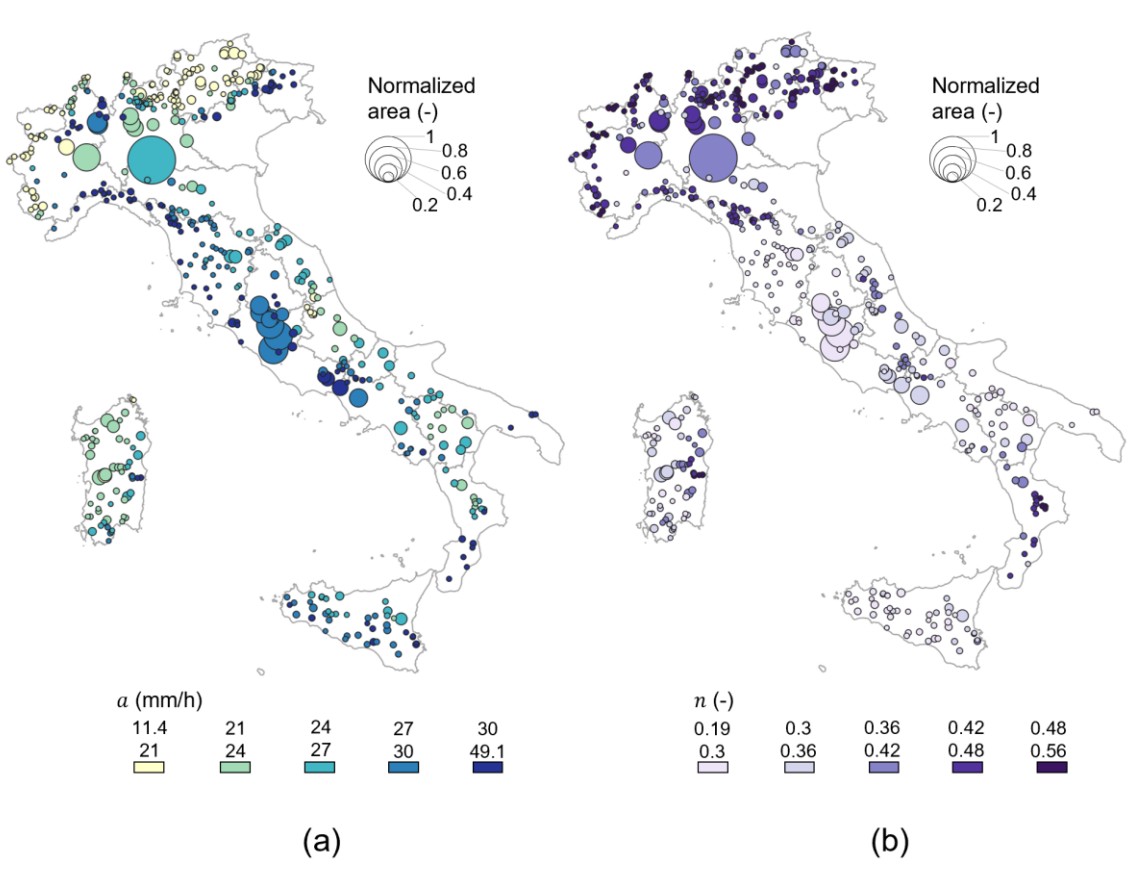

(a)             (b)

Figure 7. **Spatial representation of the mean basin parameters** $a$ **(a) and** $n$ **(b) of the IDF curves. The point size represents the basin area normalized to the maximum value within the dataset.**

## 5 Interaction between the infrastructure and the upstream basin

As mentioned in the Introduction, dams can play a significant role in mitigating the impacts of flood events, providing some level of protection to downstream areas. Well-known examples in Italy are those of the Maccheronis dam (Sardegna) which, while not specifically designed for flood attenuation purposes, significantly reduced the natural flood peak during the Cleopatra storm in 2013 (Brath, 2019). Similarly, during the Vaia storm that hit north-eastern Italy in 2018, the Ravedis (Friuli Venezia Giulia), Corlo and Pieve di Cadore (Veneto) dams played essential roles in managing floodwaters (Baruffi et al., 2019). The Chiotas and Piastra (Piemonte) reservoirs also provided assistance during a particularly intense weather event that affected the Piemonte region in October 2020 (Basano et al., 2021). These examples underscore the importance of considering unsupervised flood attenuation, i.e. based on the inherent characteristics of the landscape and the reservoir (Evangelista et al., 2023), in comprehensive flood risk management strategies.

Some basic metrics are used in the literature to roughly quantify the infrastructure's effectiveness in mitigating flood peaks. Among these metrics, one useful indicator is the relationship between the lake and the upstream basin areas (e.g. Scarrott et al., 1999; Evangelista et al., 2022), which is depicted in Figure 8a for Italian dams. All other geometric features being constant, dams with small reservoir areas and large upstream watersheds perform less effectively in flood mitigation than dams with larger reservoir areas and smaller upstream watersheds. Different values for this ratio can be viewed as a threshold below which the reservoir system is deemed to have minimal or no unsupervised attenuation effects. If adopting a ratio of 1/150, as done in Evangelista et al. (2023), one can notice from Figure 8a that approximately 50% of the Italian dams (gray points) have negligible efficiency in flood mitigation.

Another qualitative index to describe the impact of dams on floods is the ratio of the reservoir storage volume to the upstream basin area (e.g. Graf, 2006); dams with higher ratios have greater potential for flood control, as pointed out by Stecher and Herrnegger (2022). Figure 8b illustrates how these ratios are distributed throughout Italy, with particularly high values concentrated across the Alps, while the upper right scatter plot shows the relationship between the two variables.

In this paper, however, the actual values of flood accommodation capacities are provided for each Italian dam. Users can then easily and accurately assess downstream reservoir impacts without relying on the use of proxies or qualitative indicators. According to Eq. (1), the volume of water that can be stored within the reservoir for flood control purposes depends not only on its surface area, but also on a key second factor, i.e. the difference between the elevation of the maximum allowed water level and that of the spillway crest, hereinafter referred to as ΔH. The empirical distribution of ΔH values for Italian large dams is shown in Figure 8c. One can notice that 75% of the dams have a ΔH of less than 2 m, while 25% of them have a ΔH equal to 1 m. As mentioned in section 3, the higher ΔH, the greater the reservoir's capacity to effectively manage floods while maximizing its utility for hydropower generation or water storage.

In Figure 8d, the relationship between the volume allocated for flood control and the upstream catchment area is illustrated. The figure shows that the larger the basin drained by the dam, the greater the volume available for attenuating flood peaks. This pattern results from the positive correlation between lake and basin areas (with a Spearman's correlation coefficient of 0.5). The rate of increase of the available volume with basin area appears to have an upper limit, represented by the points marked in light blue in Figure 8d. This suggests that a threshold exists above which further allocation of volume for flood control would only be possible if reducing water level under the elevation of the spillway crest, and then

compromising the primary functions of hydropower generation or storage of irrigation supplies. This threshold is exceeded by only four dams (as highlighted in Figure 8d), i.e. the Lentini (Sicilia), Mamone Alaco (Calabria), Collechiavico (Lazio) and Simbirizzi (Sardegna) dams: they are located in mostly flat areas and are characterized by

comparably sized lakes and upstream watersheds, making them singular cases where the usual trade-offs between flood control and other functions are overcome.

The dam's capacity for flood control can be standardized in relation to either the reservoir size or the upstream basin area, making the information more useful than its absolute value alone. Figures 8e and 8f depict histograms of the ratio of the volume available for flood control to both the reservoir storage volume (Figure 8e) and the watershed area (Figure 8f). In

the former case, a common range of values typically falls between 0.1 and 0.5, while in the latter, values between 0.01 and 0.05 can be typically observed.

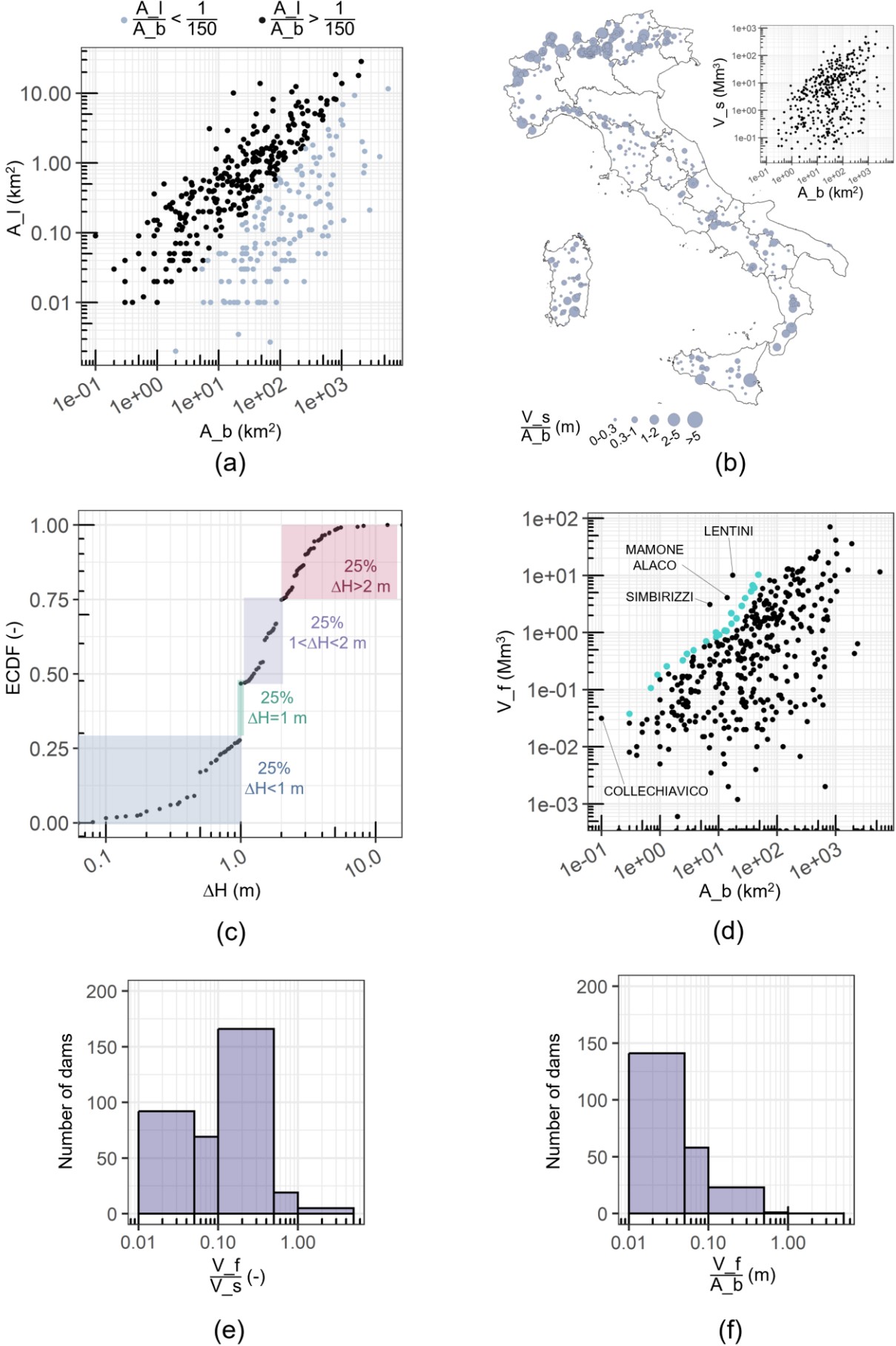

**Figure 8. Relationship between lake area (A_l) and that of the basin upstream of the dam (A_b) (a). Gray and black dots indicate dams with a ratio of lake to upstream basin areas lower or higher than 1/150, respectively. Spatial distribution of the ratio between the reservoir storage volume (V_s) and the area of the upstream catchment (A_b) (b). The scatter plot in the right corner shows the relationship between the two variables. Empirical cumulative distribution function (ECDF) of the difference between the elevation of the maximum allowed water level and that of the spillway crest (ΔH) (c). Relationship between volume available for flood control (V_f) and area of the catchment upstream of the dam (A_b) (d). Blue points represent an upper threshold for the growth rate of available volume with basin area. Histograms of the ratio of volume available for flood control (V_f) to the reservoir storage volume (V_s) (e) and to the upstream basin area (A_b) (f).**

## 6 Conclusions

In this work we provide an extensive collection of structural and catchment-related features for the full ensemble of Italian large dams which represents, to date, the most comprehensive dataset on dams in Italy. The information presented here offers a useful resource for researchers, policymakers, and stakeholders involved in water resource management and infrastructure planning.

The structural descriptors encompass both information about each dam (year of commissioning, height, type, etc.) and characteristics relating to the associated reservoir (reservoir volume and area, purposes, etc.). Unlike other similar global or national datasets, our work stands out by including information on the lake area and the elevation of the maximum allowed water level. This addition holds significant importance, particularly in estimating the dam's capacity in attenuating flood peaks.

Basin characteristics, including geomorphological, soil, land cover and climatic attributes, as well as basin contours, are determined using standardized and uniform procedures, ensuring consistency throughout the country. Taking into account the availability of the "twin" dataset from Claps et al. (2024), a wide level of detail is therefore provided on about a thousand watersheds, all over Italy, including both dammed and gauged watersheds. Given the challenges posed by climate change and water resource issues, a comprehensive analysis of our current infrastructure becomes progressively crucial. In this sense, this work can help to improve our capability to manage the complex interplay between dams and their hosting environment.

## Data availability

The dataset detailed in this paper is available at https://doi.org/10.5281/zenodo.14698223 (Evangelista et al., 2024b). It contains all the catchment boundaries and related catchment attributes described before. To access the latest version of the database, in case of future updates, the readers can refer to https://zenodo.org/doi/10.5281/zenodo.12818297 (Evangelista et al., 2024a) to download the most recent version.

## Author contribution

GE: Conceptualization, Data curation, Formal analysis, Investigation, Methodology, Software, Validation, Visualization, Writing – original draft preparation, Writing – review & editing. PM: Conceptualization, Data curation, Formal analysis, Investigation, Methodology, Software, Writing – original draft preparation, Writing – review & editing. DG: Conceptualization, Methodology, Writing – review & editing. FP: Conceptualization, Data curation, Methodology. PC:

Conceptualization, Funding acquisition, Methodology, Project administration, Resources, Supervision, Writing – review & editing.

**Competing interests**

The contact author has declared that none of the authors has any competing interests.

**Acknowledgements**

This study was carried out within the RETURN Extended Partnership and received funding from the European Union Next-GenerationEU (National Recovery and Resilience Plan – NRRP, Mission 4, Component 2, Investment 1.3 – D.D. 1243 2/8/2022, PE0000005 - Spoke TS 2).

**Appendix A**

**Table A.1 – List of secondary dams.**

| Secondary dam/s | Main dam | Region |
|---|---|---|
| Campo Moro 2 | Campo Moro 1 | Piemonte |
| Carru Segau | Medau Zimirilis | Sardegna |
| Ceresole Reale Minore | Ceresole Reale Maggiore | Piemonte |
| Cignana 2 | Cignana 1 | Piemonte |
| Colle Laura | Chiotas | Piemonte |
| Fontana Bianca Sud | Fontana Bianca Nord | Trentino Alto Adige |
| Forcoletta | Codelago | Piemonte |
| Guadalami Monte | Guadalami Valle | Sicilia |
| Lago Delio Nord | Lago Delio Sud | Lombardia |
| Lago Eugio 2 | Lago Eugio | Piemonte |
| Lago Gabiet Nord | Lago Gabiet Sud | Valle d'Aosta |
| Maria al Lago | Fedaia | Trentino Alto Adige |
| Montagna Spaccata 2 Montagna Spaccata 3 | Montagna Spaccata 1 | Abruzzo |
| Poggio Cancelli Sella Pedicate | Rio Fucino | Abruzzo |
| Rossella | Scanzano | Sicilia |
| Rio Cancello | San Eleuterio | Lazio |
| Stuetta | Cardenello | Lombardia |
| Tagliata | Giacopiane | Liguria |

**Appendix B**

**Table B.1. List of geomorphological attributes, with a brief description and an indication of the algorithm/add-on used for their computation. All the attributes are computed by processing the SRTM DEM at 30 m resolution with the *r.basin* add-on, that takes advantage of other the GRASS GIS algorithms mentioned at the beginning of section 3.3 of Claps et al. (2024). Source: Claps et al. (2024).**

| Attribute sub-category | Attribute | Notation | Units | Description |
|---|---|---|---|---|
| Altimetric and geometrical | Area | area | km$^2$ | Catchment area computed by multiplying the area of a single pixel by the number of pixels within the catchment boundary. |
| | Mean Elevation | elev_mean | m a.s.l. | Catchment mean elevation. |
| | Maximum Elevation | elev_max | m a.s.l. | Catchment maximum elevation. |
| | Minimum Elevation | elev_min | m a.s.l. | Catchment minimum elevation. |
| | Aspect | aspect | ° | Mean of the angle of exposure on the horizontal plane of each cell of the catchment. The adopted convention is that North is 0° and the aspect is computed clockwise. |
| | Hypsographic curve | elev_x | m a.s.l. | Elevation values of the hypsographic curve (i.e., the curve that defines the distribution of catchment areas located within a specific elevation range). Each $x$ corresponds to a different percentage of area (2.5, 5, 10, 25, 50, 75, 90, 95 and 97.5%). |
| | Geographic centroid | x_g y_g | m | Coordinates of the pixel nearest to the centroid of the geometric figure resulting from the projection of the catchment on the horizontal plane. |
| | Length of the orientation vector | dir_length | km | Length of the segment linking the catchment centroid to the outlet. |
| | Orientation | orient | ° | Angle of the orientation vector with respect to North. |
| | Mean slope 1 | slope1 | % | Mean slope value calculated averaging the slope map. |
| | Mean slope 2 | slope2 | % | Angle at the base of the right-angled triangle whose base is the square root of the catchment area and twice the median elevation of the catchment (relative to the closing section) as height. This slope is calculated with respect to a catchment of square shape equivalent to the real one and does not consider its actual shape, which can be elongated. |
| Horton Ratios | Horton-Strahler numbers | HS_num_u HS_length_u HS_area_u HS_slope_u | - | 4 sets of $u$ = 3 vectors (each corresponding to a Horton order), containing respectively: the number of streams of a given order [-], the average length of the streams of a given order [km], the average contributing area for each order [km$^2$] and the average slope of the streams of each order [%]. Slopes are calculated as ratio of difference of elevation between ends of the segment to its length. |
| | Area ratio | R_a | - | Ratio of the average area drained by streams of a given order $u+1$ and streams of order $u$. |
| | Bifurcation ratio | R_b | - | Ratio of number of stream branches of a given order $u$ to the number of streams branches of the next order $u+1$. |
| | Length ratio | R_l | - | Ratio of average length of streams of two adjacent orders $u$ and $u+1$. |
| | Slope ratio | R_s | - | Ratio of average slope of streams of two adjacent orders $u$ and $u+1$. |
| Streamflow network | Total stream length | TSL | km | Total length of the river network, obtained by summing the length of all its segments. |
| | Drainage density | drain_dens | km$^{-1}$ | Ratio between the total stream length and the catchment area. |
| | Length of main channel | LMC | km | Length of the longest succession of segments that connect a source to the outlet of the catchment. |
| | Length of longest drainage path | LLDP | km | Path included between the outlet and the furthest point from it, placed on the catchment boundary and identified by following the drainage directions. For most of its length the longest drainage path overlaps the main channel. |
| | Topological diameter | topo_d | - | Number of confluences found on the main channel. |
| | Mean hillslope length | MHL | km | Average of the distances (measured following the drainage directions) of all the pixels not belonging to the hydrographic network, starting from the first pixel of the hydrographic network into which they drain. |
| | Mean slope of longest drainage path | LLDP_slope | % | Calculated as $$LLDP\_slope = \frac{1}{topo\_d}\sum \frac{\Delta z_i}{L_i}\cdot 100 \qquad (B1)$$ where topo_d is the topological diameter, $L_i$ is the length of the i-th segment into which $LLDP$ is divided and $\Delta z_i$ is the corresponding elevation difference |
| Shape factor and amplitude function | Shape Factor | shape_f | - | Ratio of catchment area to the square of the length of the main channel. |
| | Elongation Ratio | elong_r | - | Ratio of the diameter of a circle of the same area as the catchment to the maximum drainage path length. |
| | Circularity Ratio | circ_r | - | Ratio between the catchment area to the area of the circle having the same circumference as the perimeter of the catchment. |
| | Compactness coefficient | c_c | - | Ratio of catchment perimeter to the diameter of the circle having the same area of the catchment. |

| | | width_mean | | Frequency distribution of the distances of each cell of the catchment, along the |
|---|---|---|---|---|
| Width function characteristics | | width_mean<br>width_var<br>width_skw<br>width_kur<br>width_x | - | Frequency distribution of the distances of each cell of the catchment, along the drainage path, to the outlet. The first four statistical moments (mean, variance, skewness and kurtosis) of this function were calculated as well as the percentiles vector containing the distance to the outlet that includes pixel percentages of x = 5%, 10%, 15%, 30%, 40%, 50%, 60%, 70%, 85% and 95%. |

**Table B.2. List of soil attributes. Source: Claps et al. (2024).**

| Attribute sub-category | Attribute | Notation | Units | Description |
|---|---|---|---|---|
| Soil | Curve Number | CN1,<br>CN1_cv<br>CN2,<br>CN2_cv,<br>CN3,<br>CN3_cv | - | Empirical parameter developed by the Soil Conservation Service (1972) and used to predict direct runoff, whose value is between 0 and 100.<br>According to the antecedent moisture condition that refers to the preceding wetness condition of soils, CN is divided into three classes, namely CN1, CN2, and CN3. CN2 is the average condition, while CN1 and CN3 represent the lowest (dry soil) and highest (saturated soil) runoff potentials, respectively. |
| | Saturated Hydraulic Conductivity | k | cm/d | Computed from sand and clay content of SoilGrids maps at 250 m resolution as follow:<br>$$k_s = 24e^{[12.012-7.55\cdot10^{-2}\,s+\frac{(-3.895+3.671\cdot10^{-2}\,s-0.1103c+8.7546\cdot10^{-4}\,c^2)}{0.332-7.251\cdot10^{-4}\,s+0.1276\log(c)}]} \quad\quad (B2)$$<br>where $s$ is the sand content (%) and $c$ is the clay content (%). |

**Table B.3. List of land cover and NDVI attributes. Source: Claps et al. (2024).**

| Attribute sub-category | Attribute | Notation | Units | Description |
|---|---|---|---|---|
| Land cover | Corine Land Cover 1 | clc1 | % | Percentage of the catchment area that corresponds to continuous and discontinuous urbanized areas (CORINE classes 111, 112). |
| | Corine Land Cover 2 | clc2 | % | Percentage of the catchment area that corresponds to woods (311, 312, 313), arboreal vegetation, shrub vegetation and bushes (CORINE classes 324, 323, 321, 322). |
| | Corine Land Cover 3 | clc3 | % | Percentage of the catchment area that corresponds to herbaceous vegetation, meadow pasture, special crops, olive groves, vineyards and arable land (CORINE classes 231, 222, 223, 221, 211, 241, 243, 242, 142). |
| | Corine Land Cover 4 | clc4 | % | Percentage of the catchment area that corresponds to non-vegetated areas (331, 333, 332, 334), mining areas, landfills, construction sites (CORINE classes 131, 133), industrial and commercial areas, and communication networks (CORINE classes 121, 122, 123, 124). |
| | Corine Land Cover 5 | clc5 | % | Percentage of the catchment area that corresponds to humid areas (CORINE classes 411, 512, 521). |
| NDVI | NDVI | NDVI<br>NDVI_cv | - | Indicator of the greenness of the biomes measured by satellite, whose value is between 0 and 1. It is defined as<br>$$NDVI = \frac{REF_{nir}-REF_{red}}{REF_{nir}+REF_{red}} \quad\quad (B3)$$<br>where $REF_{nir}$ and $REF_{red}$ are the spectral reflectances measured in the near infrared and red wavebands respectively. Mean value and (spatial) coefficient of variation were computed. |
| | B1, B2, C1, C2 | B1_NDVI,<br>B2_NDVI,<br>C1_NDVI,<br>C2_NDVI | - | Mean values of the coefficients of the Fourier series representation of NDVI (see Appendix A). |

**Table B.4. List of climatological attributes. Source: Claps et al. (2024).**

| Attribute | Notation | Units | Description |
|---|---|---|---|
| Mean Annual Precipitation | MAP<br>MAP_cv | mm | Spatial mean and coefficient of variation of the total mean annual precipitation (Braca et al., 2021). |
| B1, B2, C1, C2 | B1_rain, B2_rain,<br>C1_rain, C2_rain | - | Mean values of the coefficients of the Fourier series representative of the rainfall regime computed from the mean monthly precipitation (see Appendix A). |
| Coefficient of variation of rainfall regimes | cv_rain | - | Temporal coefficient of variation calculated from monthly mean rainfall depths derived from Braca et al. (2021). |
| Time step between maximum and minimum of mean monthly rainfall | seas_prec | - | Number of months between the occurrence of the absolute annual maximum rainfall and the subsequent absolute minimum rainfall. |
| Mean Annual Temperature | MAT<br>MAT_cv | °C | Spatial mean and coefficient of variation of the mean annual temperature computed using Braca et al. (2021). |

| Attribute | Notation | Units | Description |
|---|---|---|---|
| B1, B2, C1, C2 | B1_temp, B2_temp, C1_temp, C2_temp | - | Mean values of the coefficients of the Fourier series representation of temperature regimes (see Appendix A). |

**Table B.5. List of rainfall attributes. Source: Claps et al. (2024).**

| Attribute | Notation | Units | Description |
|---|---|---|---|
| $a$ | a<br>a_cv | mm/h | Scale factor of the IDF curve. Mean value and (spatial) coefficient of variation were computed. |
| $n$ | n<br>n_cv | - | Scaling exponent of the IDF curve. Mean value and (spatial) coefficient of variation were computed. |
| L-CV $d$h | LCV_$d$h<br>LCV_$d$h_cv | - | Coefficient of L-variation for $d$ = 1, 3, 6, 12 and 24-hour duration. Mean value and (spatial) coefficient of variation were computed. |
| L-CA $d$h | LCA_$d$h<br>LCA_$d$h_cv | - | Coefficient of L-skewness for $d$ = 1, 3, 6, 12 and 24-hours duration. Mean value and (spatial) coefficient of variation were computed. |

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
