# Peer review of "Features of the Italian Large Dams and their upstream catchments"

_Earth System Science Data, 2024_

## Author Comment (AC1)

**RC1**

The paper by Evangelista et al. introduces a detailed dataset of large dams in Italy, covering structural data (e.g., location, surface area, volume) and upstream catchment attributes (e.g., geomorphology, climate, extreme rainfall, land cover, soil).

The paper is well structured and well written, easily to follow and with figures drawn with care. The reading is smooth, and it provides detailed information regarding the methods used to build the dataset. The final aim of the dataset is of utmost importance as it may support diverse hydrological applications, from modelling to water resource management.

I have no major issues to point out, only some minor points that I list in the following.

In the Abstract "from the processing of updated and standardized grid data", please consider adding a specification of the source

*R: The abstract has been revised to be more specific.*

Add a line in between rows in the last column of Table 1 as it difficult to read, the same applies to Table 2

*R: Corrected in the revised manuscript.*

**Line 234**: "NDVI" please provide the explanation of the acronym at its first appearance

*R: Corrected in the revised manuscript.*

---

## Author Comment (AC2)

**RC2**

The MS presents the first Italian national database of large dams enriched by a wide range of structures and catchment characteristics.

The aim is to present and describe the dataset that can be used for a range of hydrological and water management issues, including flood mitigation, water resource management and so. The MS is well written, well-illustrated and pretty clear.

I really welcome the database and the overall work and I am fully convinced about the need for such an effort, whose utility is not under discussion and goes beyond the open access policy of environmental data.

I also encourage the authors to pursue this type of initiatives, hoping Italy will have in the future a database similar to the French ROE, which includes the whole range of river infrastructures at national scale, not only hydrologically relevant ones (see: https://www.eaufrance.fr/les-obstacles-lecoulement-des-eaux-de-surface).

I only have minor concerns:

- Some recent and relevant publications on the topic are missing, notably: Belletti et al. 2020, which already published the location of such dams within a larger dataset of man-made barriers concerning European rivers in the context of the H2020 AMBER project (https://amber.international/). As well, the authors should probably mention the recent GDW initiative that, building on previous works and existing networks, is trying to build such detailed database at the global scale (https://www.globaldamwatch.org/).

  *R: We thank the reviewer for this comment and have updated the list of references to include recent, relevant publications on global and continental dataset construction initiatives. Specifically, the works of Belletti et al. (2020), Paredes-Beltran et al. (2021), Lehner et al. (2024) have been included.*

- The definition of the structures included in the database is incomplete: what do you mean with river barriers? I would suggest to add pictures and description for the different dam types (as an annex). This is needed to make the MS and the database accessible to a wider public (e.g. not specialists in hydrology or dam engineers).

  *R: We thank the reviewer for pointing out this potential source of misunderstanding. According to the Italian regulations (Decree of the Minister of Public Works, March 24, 1982 - Technical standards for the design and construction of dam barriers), a "river barrier" is defined as a structure designed to create a contained backflow within the riverbed, primarily aimed at raising the water level upstream to facilitate water diversion and, more generally, at regulating water levels. Consequently, a river barrage is not typically intended for water storage. On the other hand, a "dam" is intended as a structure that intercepts a watercourse and causes the temporary accumulation of water in the valley upstream of the dam for purposes such as hydroelectric power generation, irrigation, industrial use, or other human needs. This distinction is also applied in the technical literature. We have clarified this difference in the revised version of the manuscript.*
  *However, we recognize that the term "river barrier" might be confusing, as it is often interpreted in other works (e.g. Belletti et al., 2020, Lehner et al., 2024) to refer to any obstruction to natural river flow. To avoid this ambiguity, we replaced "river barrier" (with the meaning clarified above) with the more widely accepted "river barrage" (e.g. Speckhann et al., 2021, Hydraulic Engineering of Dams, Hager et al., 2020). We will also apply this revision to the dataset uploaded to Zenodo.*
  *We believe that including explanatory photographs for each type of structure may not be necessary. The terminology used (i.e. that used in Figure 1c) is based on widely recognized technical literature, which is readily accessible to all. By using these terms, readers can easily find relevant images and detailed descriptions online, such as on the ICOLD website.*

- The aim of the work is unclear times to times: is it the publication of the database? the characteristic provided? the new characteristics measured? The indicators/metrics? Please clarify.

*R: In line with the journal's standards, our work aims to provide a comprehensive and easily accessible dataset to support scientists and practitioners in their research and applications. To achieve this, we have focused on two key points:*

    *i.    Some of the data included in this dataset, such as dam locations, their height or storage volume, were previously available through various sources; however, these data were often fragmented, which made them difficult to use effectively. By standardizing this information, we offer a more consistent and user-friendly resource.*

    *ii.    In addition to integrating existing data, we have introduced new characteristics that were not previously available. For instance, we calculated a large number of upstream basin attributes, which add valuable information and enhance the dataset's utility. It is worth mentioning that this dataset is consistent with the recently released Italian FlOod and Catchment Atlas (FOCA, https://doi.org/10.5194/essd-16-1503-2024), as they share the same computed basin descriptors and knowledge bases. FOCA is a dataset of gauged catchments. Therefore, this shared knowledge can be used for modelling different hydrological and water-related processes.*

*Furthermore, the metrics presented in Section 5, while not part of the dataset itself, are intended to illustrate one possible application of the data, i.e. the interaction between the dam infrastructure and its upstream basin, showcasing the practical value of this dataset.*

- Finally, the scope of the database seems a bit too narrow, notably in the discussion part (chapter 5): the authors mainly highlight flood mitigation issues but they could have mentioned broad water resources management issues (e.g. water scarcity). Some critical considerations about environmental challenges related to the presence of dams versus other (human) uses would have been welcomed as well. Actually, it is a pity that the authors don't further discuss the catchment scale characteristics and how these can be used for different purposes.

*R: We agree with the reviewer that expanding the discussion to include broader water resource management issues, such as water scarcity and environmental impacts, would provide a more comprehensive perspective. While we acknowledge the relevance of these topics, addressing them would require additional information that encompass aspects like water demand, usage patterns, and environmental health indicators, which are not included in the current dataset.*

*For instance, to assess water scarcity, additional data on regional water demand, local climate conditions (e.g., temperature, humidity, wind speed), or water use by different sectors (agriculture, industry, etc.) would be essential. Similarly, evaluating the environmental challenges posed by dams—particularly in terms of their interaction with downstream ecosystems—would require data on biodiversity, water quality, sediment transport, and other ecological health indicators. Nevertheless, the availability of catchment boundaries in our dataset allows users to compute catchment-averaged values for other parameters, such as evapotranspiration and snow cover, using their own models or alternative datasets.*

*Given these limitations, the discussion section of the manuscript primarily focuses on the interaction between the dam infrastructure and the upstream basin.*

*To be consistent, the old sentence in the Abstract "This dataset is expected to be of great help for a broad spectrum of hydrological applications, ranging from modelling to water resource management"' has now been revised to: "This dataset is expected to be of great help for a broad spectrum of hydrological applications, particularly those related to floods".*

I have provided some comments and suggestions in an edited version of the PDF, see attached.

Comments from the attached PDF:

**Line 24**: see also: https://www.pnas.org/content/117/7/3648 https://www.nature.com/articles/s41467-023-40922-6 10.1038/nature09440

*R: As suggested by the reviewer, we have added additional references.*

**Line 25**: and also on the management at catchment scale see also: http://www.sciencedirect.com/science/article/pii/S0048969717334198 http://wp.iwaponline.com/content/11/s1/121

*R: As suggested by the reviewer, we have added an additional reference.*

**Line 27**: I guess you can mention that many large dams have been built in the 19s and 20s and are getting old and in need for maintenance

*R: The text has been revised as suggested by the reviewer.*

**Line 31**: their age, state of use (and their purpose)

*R: The text has been revised as suggested by the reviewer.*

**Line 33**: see also Global Dam Watch initiative and cited databases/references:  https://www.globaldamwatch.org/

*R: R: Additional references on global and continental dataset construction initiatives have been included (please refer to the comment above).*

**Line 36**: see also v2 within the GDW consensus db

*R: Additional references on global and continental dataset construction initiatives have been included (please refer to the comment above).*

**Line 43**: the DEM is from SRTM

*R: We corrected it in the revised version of the manuscript.*

**Line 46**: what about other global db? (see GDW website)

*R: Following the reviewer's suggestions, the reference list on large-scale datasets has been expanded.*

**Line 47**: unclear?

*R: The sentence has been rephrased in the revised manuscript to be clearer.*

**Line 49**: check the new DB

*R: We checked the number of dam points in Italy included in the new GDW dataset. This information has been added to the revised manuscript.*

**Line 52**: could you clarify what do you intend here?

*R: The sentence has been rephrased in the revised manuscript to be clearer.*

**Line 56**: this is true, but you should welcome such regional (see AMBER) or global (such GDW) initiatives and state that your work is really valuable to feed them...

*R: We agree with the reviewer and have added a comment about this in the revised manuscript.*

**Line 58**: actually the process is the contrary.... national or more often local datasets exist and the regional or global initiatives are intended to give a global picture, and to support locally when data are missing at local sale...
Your paper/dataset is welcome because:
(i) we were extremely in late in Italy (national register of dams existed but data wasn't made available)
(ii) such kind of initiative can help to better feed global scale datasets that are indeed incomplete (because of lack or uncompleteness of national/regional db)

*R: We agree with the reviewer on the ambiguity of the sentence "To fill these gaps, the production of nationwide datasets was undertaken" and have decided to remove it from the revised manuscript.*

**Line 59**: amongst others... see GDW website and AMBER project with inventory of existing national db in EU

*R: Following the reviewer's suggestions, the reference list on large-scale datasets has been expanded.*

**Line 65**: for doing what? this is not clarified above

*R: We have clarified this point in the revised manuscript.*

**Line 76**: a definition of large dams should be provided

*R: This definition is given in Line 51 in the original manuscript.*

**Line 91**: that means these fully transparent? (i.e. no water storage/lake). could you clarify "being tested"?

*R: In the revised version of the manuscript we have clarified both these points.*

**Line 94**: can you define "river barriers"? Does it correspond to the definition given below (structure located at large natural lakes)

*R: A clearer definition of "river barrier" has been provided in the revised manuscript (please refer to the comment above).*

**Line 102**: it should be provided a definition of the different types (a Table?)

*R: Please refer to the comment above.*

**Line 106**: manually? please clarify

*R: This check was done manually. We have specified it in the revised version of the manuscript.*

**Line 110**: Have other dams been built in Italy after the last in 2010?

*R: The Pratolungo Dam, located in the Lazio region, is the only dam whose construction began after 2010. Its construction is still ongoing. In the revised manuscript we have clarified that Figure 4 excludes dams currently under construction and those for which the start and end dates of construction are not available.*

**Line 113**: and to assess (plan) maintenance

*R: While the end year of construction of dams is certainly important for planning maintenance, the line mentioned by the reviewer pertains specifically to the hydrological applications that requires this information. To maintain consistency, we have decided not to incorporate this comment.*

**Line 117**: Y axis unit is not the number but proportion, right? Please specify

*R: Yes, it is correct. We have specified it in the revised manuscript.*

**Line 119**: this information should go in the main text

*R: The content the reviewer is referring to is already included in the main text, as indicated by its lack of bold formatting. To avoid any potential misunderstanding, we have added a space between the end of the figure caption and the beginning of the main text.*

**Line 125**: all these are from the GDD?

*R: All structural features in Table 2, except for the reservoir volume available for flood attenuation ($V\_f$), were provided directly by the General Department of Dams and Hydro-Electrical Infrastructures (GDD). $V\_f$ was calculated using Eq. 1, with input data sourced from the official GDD records (after validation in the case of the lake area). This is specified in lines 134-136 of the original manuscript.*

**Line 126**: but see GDW

*R: A comment has been included about the availability of reservoir surface areas in the GDW dataset.*

**Line 136**: is this originally sourced from GDD?

*R: The volume available for flood mitigation is not directly provided by the GDD. Rather, we estimated it using Eq. (1) and data sourced from the GDD, i.e. the elevation of the maximum allowed water level, the elevation of the spillway crest and the lake area.*

**Line 148**: so, why did you validated the lake area?

*R: Errors in the value of the lake area within the official records may have occurred during the compilation of the GDD dataset, as inconsistencies can arise in large-scale data processing. While such errors could potentially affect other structural characteristics as well, the reservoir surface area is a parameter that can be verified, for example, by using a high-resolution digital elevation model, as we did in our work. On the other hand, verifying other geometrical characteristics is more challenging. Furthermore, the lake area is critical for a variety of technical applications, including hydrological modelling, flood risk assessment, and water volume estimation. It is also essential for evaluating the dam's capacity to attenuate flood peaks, as discussed in the manuscript.  Given the significant role the lake area plays in these applications and its direct influence on the accuracy of hydrological models, we deemed it appropriate to validate this value.*

**Line 153**: Figure 3 not mentioned in the text

*R: We have commented Figure 3 in the text in the revised manuscript.*

**Line 158**: agree! but what is the link with the MS?

*R: We believe it is important to emphasize the linear relationship between lake area and storage volume, as shown in the Figure the reviewer is referring to. This relationship can be applied in a variety of practical contexts. For example, it can be used to design synthetic dams with a realistic and feasible lake area/storage volume ratio. This is particularly useful for simulations, where the goal is to evaluate the performance of hypothetical dams under different conditions. Additionally, this relationship could be beneficial in other areas such as preliminary design stages.*

**Line 189**: at which resolution?

*R: We have specified the spatial resolution of the DEM in the revised manuscript.*

**Line 196**: described in

*R: This error has been corrected.*

**Line 204**: missing ref.

*R: Added in the revised manuscript. We thank the reviewer for having pointed out this oversight.*

**Line 212**: clarify

*R: We have provided a more detailed explanation in the revised manuscript.*

**Line 217**: yes but you validated the data, as mentioned in the paragraph below - please rephrase

*R: The sentence has been rephrased to be more consistent.*

**Line 235**: check

*R: Following the reviewer's suggestions, we have included tables describing all computed basin attributes in Appendix B of the revised manuscript.*

**Line 245**: you should list the attributes at least. Even if the method is the same than Claps et al., the MS should be self-explaining

*R: Our initial decision to exclude the list of basin attributes from the manuscript was aimed at avoiding potential plagiarism issues, as the tables describing the meaning of each attribute are identical to Tables from 1 to 5 of Claps et al. (2024). However, we appreciate the reviewer's suggestion and have now included them in Appendix B in the revised manuscript.*

**Line 253**: in case of basin transfers? please clarify

*R: We have clarified the meaning of indirectly connected basin in the revised manuscript.*

**Line 260**: why the only benchmark? I would explain in one sentence or two the nature/content of this database, also because the report is in Italian (the link to the report doesn't work)

*R: The only other source available for comparing basin area values is the report published by the Istituto Superiore per la Protezione e la Ricerca Ambientale (ISPRA) (Policicchio et al., 2020). Indeed, as mentioned in the Introduction, neither the General Department of Dams nor any other source provides information on the upstream catchments of Italian dams. We agree with the reviewer that providing additional context about this report would be beneficial for non-Italian speakers. We have included a brief description in the text. We also appreciate the reviewer highlighting the issue with the link, which has now been replaced.*

**Line 263**: can you check this?

*R: Unfortunately, this information cannot be directly verified, as the report by Policicchio et al. (2020) provides no details on how catchment boundaries were delineated. The discrepancies we observed in basin areas compared to those reported (see Figure 6a) may stem from several factors. One likely explanation is inaccuracies in the placement of outlet coordinates. Even slight errors in positioning, especially at river junctions, can significantly impact the size of the delineated catchment area. Another possibility is the inclusion of indirectly connected basins in the upstream area computation, regions that, while not natural contributing areas, may be hydrologically linked through artificial connections, leading to discrepancies in the computed basin area. Additionally, the use of a digital elevation model (DEM) with a different spatial resolution (again not specified in in Policicchio (2020)) may have further contributed to these differences, particularly in smaller catchment areas where spatial resolution plays a more critical role.*

**Line 283**: same comment here: you should provide at least the list here (as Annex?)

*R: Please refer to the comment above.*

**Line 292**: about what? Carriero (2004) missing ref.

*R: We have clarified in the revised manuscript that the maps we refer to are Curve Number maps and have included the missing reference. We thank the reviewer for having pointed out this oversight.*

**Line 294**: it's really curious that you criticize global db for dams while you use global db for soil data

*R: We believe that global datasets may not be the most suitable option for working at a national scale. However, in the case of Italy, there is currently no national, commonly recognized dataset on soil properties. While geological data is available, soil-specific data is more limited. Some European datasets were available when we started working on this manuscript. One of the most used is the LUCAS (Land Use/Land Cover Area Frame Survey) dataset. Nevertheless, we chose to use SoilGrids for the following reasons:*

  *i.    SoilGrids incorporates the sampling points used in LUCAS and integrates them with additional data from the WoSIS (World Soil Information Service) points.*
  *ii.   The $R^2$ for particle size fractions, which we used here to estimate saturated hydraulic conductivity, is slightly higher in SoilGrids (ranging from 0.6 to 0.65) compared to LUCAS (ranging from 0.47 to 0.5).*
  *iii.  LUCAS only provides data for the top 20 cm of soil, whereas SoilGrids offers data up to a depth of 2 meters, which aligns better with our interest in the 30 cm depth, as explained in the manuscript.*

*Additionally, the spatial resolution of the SoilGrids maps (250 m) further influenced our decision to choose this dataset.*

**Line 307**: what do you mean by "consistency of vegetation"?

*R: The sentence has been rephrased in the revised manuscript to be clearer.*

**Line 329**: as well, it would be useful to see the list of these attributes

*R: Please refer to the comment above.*

**Line 356**: Again, it would be useful to see that list

*R: Please refer to the comment above.*

**Line 383**: I guess the ratio is not the only relevant parameter here. What do you do with all the other parameters calculated above at catchment scale? It would have been interesting to quantify/estimate the effects of the other parameters calculated at catchment scale on the variability of this ratio (the scatterplot shows high variability in the low residuals also within the two sub-populations above/below 1/150, which would deserve to be further explored and explained). The message that is passing here is that dams would solve all flood issues and I'm sure it's not the case.

As well, assessing the only positive effects of dams on floods without discussing about the negative effects is telling only one part of the story. If I'm not wrong the aim of the work is to present and describe the dataset that can be used for a range of hydrological and water resource management issues, and not to sell flood mitigation properties of dams.

*R: As mentioned in a previous comment, the metrics presented in Section 5, such as the ratio between the lake and the catchment area, are intended to illustrate one possible application of the data, i.e. the interaction between dam infrastructure and its upstream basin. In an earlier comment, we clarified that different applications related for instance to water scarcity or environmental issues would require additional data, which is not included in the current version of the dataset. Following the reviewer's suggestions, we have rephrased part of the text in the revised manuscript to avoid any potential misunderstanding.*

*The inherent variability in the relationship between the catchment and reservoir surface areas, as observed by the reviewer in Figure 8a, is shaped by the specific topography and geographical characteristics of the region where the dam is situated. This relationship is site-specific and cannot be readily generalized across different regions or dam locations. For these reasons, we believe that attempting to assess this variability based on the basin attributes we calculated would not provide meaningful insights.*

*Regarding flood mitigation, it is well-established that dams can play a relevant role in reducing the impacts of floods, even when active gate operations are not employed (i.e. when unsupervised attenuation is performed). The ratio between the lake and the catchment area is one of the indicator that can be used for estimating a dam's potential for flood mitigation, as it provides a measure of the reservoir's potential to accommodate and manage inflows during flood events. The same applies to the ratio between the storage volume and the upstream basin area. However, investigating these characteristics does not imply that dams can solve any issues related to floods and this is not the message we want to communicate. The mitigation capacity, whether supervised or not, is a highly relevant topic for the scientific community, and we believe that the observations and figures presented in Section 5 can contribute to providing a comprehensive overview of the characteristics of Italian dams with regard to this topic.*

**Line 387**: same comment as on Figure 8a

*R: Please refer to the comment above.*